# Metabolomic Profiling and Network Toxicology: Mechanistic Insights into Effect of Gossypol Acetate Isomers in Uterine Fibroids and Liver Injury

**DOI:** 10.3390/ph17101363

**Published:** 2024-10-12

**Authors:** Zishuo Liu, Hui Zhang, Jun Yao

**Affiliations:** 1School of Pharmacy, Xinjiang Medical University, Urumqi 830017, China; liuzishuo@stu.xjmu.edu.cn (Z.L.); xydzhanghui@stu.xjmu.edu.cn (H.Z.); 2Key Laboratory of Active Components and Drug Release Technology of Natural Medicines in Xinjiang, Xinjiang Medical University, Urumqi 830017, China

**Keywords:** gossypol optical isomer, serum metabolomics, network toxicology

## Abstract

Objective: Gossypol is a natural polyphenolic dialdehyde product that is primarily isolated from cottonseed. It is a racemized mixture of (−)-gossypol and (+)-gossypol that has anti-infection, antimalarial, antiviral, antifertility, antitumor and antioxidant activities, among others. Gossypol optical isomers have been reported to differ in their biological activities and toxic effects. Method: In this study, we performed a metabolomics analysis of rat serum using 1H-NMR technology to investigate gossypol optical isomers’ mechanism of action on uterine fibroids. Network toxicology was used to explore the mechanism of the liver injury caused by gossypol optical isomers. SD rats were randomly divided into a normal control group; model control group; a drug-positive group (compound gossypol acetate tablets); high-, medium- and low-dose (−)-gossypol acetate groups; and high-, medium- and low-dose (+)-gossypol acetate groups. Result: Serum metabolomics showed that gossypol optical isomers’ pharmacodynamic effect on rats’ uterine fibroids affected their lactic acid, cholesterol, leucine, alanine, glutamate, glutamine, arginine, proline, glucose, etc. According to network toxicology, the targets of the liver injury caused by gossypol optical isomers included HSP90AA1, SRC, MAPK1, AKT1, EGFR, BCL2, CASP3, etc. KEGG enrichment showed that the toxicity mechanism may be related to pathways active in cancer, such as the PPAR signaling pathway, glycolysis/glycolysis gluconeogenesis, Th17 cell differentiation, and 91 other closely related signaling pathways. Conclusions: (−)-gossypol acetate and (+)-gossypol acetate play positive roles in the treatment and prevention of uterine fibroids. Gossypol optical isomers cause liver damage through multiple targets and pathways.

## 1. Introduction

Uterine leiomyoma (UL) is a hormone-dependent benign tumor that occurs in the female myometrium and is most common in women aged 30–50 years, with women who are around 50 years old and near menopause representing the peak age of the onset of this disease [1]. Most patients in the early stage of uterine fibroid formation are asymptomatic, as these are often only found incidentally during pelvic or ultrasound examinations. Due to the production of tumor masses in the uterine tissue, the uterine cavity and adjacent organs (the ovaries, fallopian tubes) are compressed, resulting in uterine bleeding, infertility, ectopic pregnancies, spontaneous abortion, anemia and other clinical manifestations [2]. When fibroids grow, they mostly protrude from the serosal surface of the uterus, and the blood vessels connected to the tumor mass can easily be twisted and torn, causing severe acute pain in the abdomen [3]. It is generally believed that the abnormal secretion of sex hormones such as estrogen and progesterone is the main reason for the production and growth of uterine fibroids, which can promote mitosis, the proliferation of smooth muscle cells and the growth of leiomyomas [4,5]. Uterine fibroids are benign tumors with a malignancy rate of 0.4% to 1.25% and are usually associated with a risk of fibroid progression when they grow rapidly in the short term [6]. Therefore, reducing fibroid cell proliferation, shrinking the volume of uterine fibroids and removing tumors are the primary therapeutic goals for uterine fibroids.

Uterine fibroids are mainly treated via surgery or the administration of drugs. The former can directly remove the tumor; however, it causes significant trauma and the recurrence rate after surgery is high [2]. In terms of drug treatments, the most commonly used drug is the anti-progestin preparation mifepristone. Another drug compound listed as suitable for treating uterine fibroids is gossypol acetate, which is a non-hormonal drug administered in the form of a tablet, and can effectively inhibit the secretion of steroidal hormone receptors in the uterine smooth muscle, endometrium and other parts of the uterus. It leads to a reduction in or elimination of fibroids and plays a role in treating and relieving symptoms of uterine fibroids. But the elevated hepatic SGPT level and GSH content were induced by mixed gossypol. Different doses of mixed gossypol acetate can cause significant pathological changes in rat liver tissue, such as mitochondrial vacuolization, endoplasmic reticulum dilation, perinuclear space widening and glial fiber proliferation in Disse space. Meanwhile, mixed gossypol acetate produced a large amount of O_2_ and H_2_O_2_, which affected the binding of liver to microsomal proteins [7,8]. The effect of mixed gossypol on the liver is revealed at the pathological level.

Because gossypol is a racemate and a natural product of polyphenol bisaldehyde that is primarily isolated from cottonseed [9,10], it has a variety of biological activities, including anti-infective [9], antimalarial [11], antiviral [12], antifertility [7], antitumor [13] and antioxidant activities [14]. The biological activity and toxic effects of (+)-gossypol and (−)-gossypol have been reported to differ [15,16]. The antitumor effect of (−)-GA is more potent than that of its racemate (+)-GA [17,18,19,20]. However, (+)-gossypol has a stronger destructive effect on DNA bonds in normal human leukocytes than (−)-gossypol [16].

The differences in efficacy and adverse effects of gossypol optical isomers have been elucidated in previous studies [21]. To further clarify the mechanism of the effect of acetate gossypol optical isomers on uterine fibroids and liver injury, 1H-NMR technology was used to perform metabolomics analysis of rat serum to investigate the mechanism of gossypol optical isomers on uterine fibroids. It was found that gossypol could significantly improve the abnormality of tricarboxylic acid cycle, immune function, glycolysis and gluconeogenesis metabolism caused by uterine fibroids. Network toxicology was used to explore the mechanism of liver injury caused by acetate gossypol optical isomers. The potential targets of liver injury (HSP90AA1, HSP90AB1, SRC, MAPK1, AKT1, EGFR, BCL2, CASP3) and the molecular mechanisms of liver injury in rats (cancer pathway, PPAR signaling pathway, gluconeogenesis/glycolysis and Th17 cell differentiation) were elucidated. It has provided a research basis and reference for the further exploration of the drug use of optical isomers.

## 2. Results

### 2.1. Untargeted Serum Metabolomics

#### 2.1.1. Analysis and Attribution of Rat Serum’s 1H-NMR Metabolite Profiles

As shown in Figure 1A, we have generated attribution maps of rat serum from the normal control group, the uterine fibroid model group, the high-dose (+)-gossypol acetate group, the high-dose (–)-gossypol acetate group and the positive control group. The 1H-NMR spectra of these five groups’ rat serum were segmented, and the resulting integral values were analyzed via PLS-DA to obtain spatial distribution maps (3D plots) (Figure 1B). In the PLS-DA analysis, R^2^X = 0.452, R^2^Y = 0.331, Q^2^ = 0.263 and each group occupies an independent space, indicating that each group’s serum has different metabolic components.

#### 2.1.2. Analysis of Rat Serum OPLS-DA Results

The OPLS-DA mode was used to compare the serum of rats in the model group of uterine fibroids with the normal control group, while the rats in the drug groups and rats in the model group were compared and analyzed to determine the differential metabolites between the two groups and analyze their differences. As can be seen from Figure 2, the distribution of each group in the two comparisons is completely separated, indicating that the serum of the two groups of rats has obvious differences in its metabolic components. In this analysis, the distribution of the metabolites in the normal control group and the model group is completely separate: R^2^X = 0.374, R^2^Y = 0.917, Q^2^ = 0.63 (Figure 2A). R^2^X = 0.435, R^2^Y = 0.978, Q^2^ = 0.831 are the values seen for the positive control group versus the model group (Figure 2B), while the high-dose (+)-gossypol acetate group and model group generate R^2^X = 0.271, R^2^Y = 0.928, Q^2^ = 0.536 (Figure 2C). Furthermore, these values were R^2^X = 0.524, R^2^Y = 0.873, and Q^2^ = 0.63 for the comparison of the high-dose (−)-gossypol acetate group and the model group (Figure 2D).

#### 2.1.3. Statistics and Analysis of Serum Metabolic Differentiators in All Groups of Rats

The differential metabolic components in the serum of all groups of rats were statistically obtained from nuclear magnetic resonance hydrogen spectroscopy and then summarized and tabulated. The results showed that there were significant differences between the serum of rats in the positive control group, the high-dose (+)-gossypol acetate group, the high-dose (–)-gossypol acetate group, the normal control group and the uterine fibroid model group. When the correlation coefficient r > 0 (or r < 0), it indicates that there is a difference in the content of a metabolite between two groups and that the content in one group shows a decreasing (or increasing) trend.

Elevated levels of lysine, glutamic acid, alpha-aminobutyric acid, lactic acid, unsaturated fatty acids, proline and glycine were seen in the serum of blank control rats compared to that of the uterine fibroid model group, while their levels of arginine, β-glucose, citrulline, α-glucose and glycerol decreased (*p* < 0.05). The levels of lactic acid, glutamic acid, urea and formic acid in the serum of the high-dose (+)-gossypol acetate group were increased, while its lipid, proline, β-glucose and α-glucose contents were decreased (*p* < 0.05). The levels of cholesterol, glutamate, lipids, acetoacetate, lactate and unsaturated fatty acids were significantly higher in the differential metabolites of the serum from the high-dose (–)-gossypol acetate group, while its glutamine, proline, α-glucose, β-glucose, citrulline and glycerol levels were significantly decreased (*p* < 0.05). The levels of cholesterol, lactic acid, glutamic acid, pyruvate and unsaturated fatty acids in the serum of the positive control group increased, while the levels of arginine, proline, α-glucose, β-glucose, citrulline and glycerol decreased (*p* < 0.05). The data on the chemical shifts in the differential metabolites and their attributions and correlation coefficients are detailed in Table 1 and Table 2.

**Table 1 pharmaceuticals-17-01363-t001:** Comparison of correlation coefficients of major serum metabolites in each group (*n* = 8).

Serial Number	Metabolites	Comparison of Normal Control and Model Groups	Comparison of (+)-Gossypol Acetate and Model Groups	Comparison of (−)-Gossypol Acetate and Model Groups	Comparison of Positive Control and Model Groups
1	Cholesterol	−0.60	0.41	−0.78	−0.69
2	Isoleucine	−0.54	—	—	−0.38
3	Leucine	—	0.55	—	—
4	Lipid	−0.61	0.69	−0.80	−0.52
5	Lactic acid	−0.69	−0.84	−0.74	−0.88
6	Alanine	−0.49	−0.23	0.40	—
7	Acetic acid	—	—	—	−0.47
8	Lysine	−0.78	—	—	—
9	Glutamic acid	−0.74	−0.65	−0.76	−0.63
10	Methionine	—	—	—	−0.37
11	Acetoacetate	−0.59	0.43	−0.52	−0.38
12	Glutamine	—	0.49	0.67	—
13	Pyruvic acid	−0.42	−0.45	—	−0.67
14	Citrate	—	−0.52	—	—
15	γ-aminobutyric Acid	−0.77	—	0.56	—
16	Choline	−0.48	−0.60	—	−0.57
17	Arginine	0.73	—	0.62	0.81
18	Proline	−0.77	0.78	0.75	0.92
19	β-glucose	0.90	0.98	0.99	0.96
20	α-glucose	0.84	0.71	0.90	0.95
21	Glycine	−0.67	0.60	—	—
22	Citrulline	0.84	0.61	0.90	0.86
23	Glycerin	0.74	0.47	0.89	0.89
24	Creatine	−0.38	—	—	—
25	Unsaturated Fatty acids	−0.70	−0.33	−0.87	−0.71
26	Urea	—	−0.77	—	—
27	Formic acid	—	−0.35	—	—

Note: The metabolites with positive correlation coefficients in this table are metabolites with a reduced content compared to the model group. Metabolites with a negative correlation coefficient are those with an increased content compared to the model group, while “—” indicates no change.

### 2.2. Network Toxicology Analysis

#### 2.2.1. Prediction of Targets

A total of 569 target components of gossypol acetate were predicted by Pharm Mapper and the Swiss Target Prediction platform. After removing repeated targets, a total of 301 target components were obtained. A total of 190 hepatotoxic targets related to gossypol optical isomers were screened by drawing on the cross-section of a Venn diagram (Figure 3A).

#### 2.2.2. Construction and Analysis of Protein Interaction Networks

The 190 targets acquired were imported into the STRING database in order to download the results of their protein interactions; the confidence level was set to 0.700, the discrete nodes were hidden and the results were imported into Cytoscape software 3.9.1 to plot their PPI networks (Figure 3B). The degree of inter-target interactions varies, with nodes denoting proteins and edges denoting inter-protein associations. Node sizes and color depths are positively correlated with their degree value. The larger the degree value, the higher the score and the more critical the role of the target. The network graph has 175 nodes and 730 edges; its average node degree is 7.64 and its average local clustering coefficient is 0.433.

#### 2.2.3. GO Gene Function and KEGG Pathway Analyses

The results of the GO analysis included three branches: biological processes, molecular functions and cellular components (Figure 3C). There were 474 results for biological processes (BPs), which mainly involved the “response to hormones”, “protein phosphorylation”, “positive regulation of phosphorylation” and other processes. A total of 28 results were enriched cell components (CCs), and these mainly involved the “vesicle cavity”, “receptor complex”, “side of membrane”, etc. A total of 35 results were obtained from the molecular function (MF) analysis, and these mainly included “phosphotransferase activity, receptor alcohol group”, “kinase binding”, “oxygen multireductase activity” and other functions.

A total of 91 pathways were enriched in the KEGG pathway analysis. The key pathways included pathways related to cancer, the PPAR signaling pathway, glycolysis/gluconeogenesis and Th17 cell differentiation (Figure 3D).

## 3. Discussion

### 3.1. Analysis of the Mechanism of Action of Gossypol Optical Isomers on Serum Metabolomics in Rats with Uterine Fibroids

The preliminary study of 1H-NMR metabolomics found that there were differences in the metabolites in the serum of rats from the uterine fibroid model group and the normal control group. The serum levels of isoleucine, alanine, lysine, glutamic acid, glycine, proline, acetoacetate, choline, pyruvate, lactic acid, γ-aminobutyric acid, lipids (including LDL and VLDL), cholesterol, creatine, unsaturated fatty acids and other metabolites in the model group were significantly decreased, while the levels of glucose, glycerol, arginine and citrulline were increased. The above metabolites constitute the metabolomic features of the uterine fibroid model rat.

The changes in the levels of various amino acids in the uterine fibroid model group indicated that uterine fibroids disrupt amino acid metabolism. The essential amino acids isoleucine, leucine, alanine and lysine are branched-chain amino acids that participate in the synthesis and decomposition of proteins in the body [22]. Amino acids can also be used as energy sources to meet the energy needs of the body and provide energy regulation, thus maintaining the nitrogen balance of the body [23]. Alanine is very important for cell growth and physiological metabolism. It is one of the most important amino acids that make up proteins [24]. Arginine, glutamate and proline can be combined with alanine to form glutamine, which can be deaminated to α-ketoglutarate and NH4+ by glutamine dehydrogenase, or deaminated to α-ketoglutarate by alanine transaminase. α-ketoglutarate then enters the tricarboxylic acid cycle to meet the energy needs of the body. Glutamine, glutamate and other metabolites play an important role in maintaining the normal immune function of the body; glutamine and glutamate, as precursors of the synthesis of the natural antioxidant glutathione (GSH), have an important antioxidant effect on the body’s cells [25]. Glutamine and creatine are also basic metabolites that maintain the normal structure of cells. The changes in the levels of these serum metabolites in the uterine fibroid model group reflect a disorder of amino acid metabolism in the body, which causes abnormal energy metabolism and weakened immune function, suggesting that uterine fibroids may cause damage to the body’s immune function in vivo.

The significant increase in serum glucose and the decrease in lactic acid seen in this group may be due to abnormal changes in the glycolysis metabolism process. When the body’s glucose content increases, its cells undergo aerobic oxidation to carry out glycolysis, releasing a large amount of lactic acid in the process [26]. However, when the amount of glucose in the body is increased, the aerobic oxidation occurring in the cells is abnormal and glycolysis is not carried out in time, resulting in a significant reduction in the content of lactic acid, the product of the glycolysis process, indicating that the body’s glycolytic metabolism is abnormal. Pyruvate and lactic acid are important intermediates in gluconeogenesis, and pyruvate can be converted to lactic acid under the catalysis of related enzymes. However, the serum levels of pyruvate and lactate were reduced in the model group, suggesting that these rats’ gluconeogenesis was abnormal. At the same time, pyruvate and lactate are also the key intermediates of energy metabolism (the tricarboxylic acid cycle); when their serum levels are reduced, as in the model group, this slows down the body’s fatty acid β-oxidation, which in turn causes a decrease in the body’s levels of lipids, cholesterol and unsaturated fatty acids and an increase in its level of glycerol, suggesting a disturbance in the body’s energy metabolism.

A comparison of the differential metabolites in the serum of the groups administered gossypol acetate and that of the model group revealed a trend toward higher levels of lactate, cholesterol, leucine, alanine and glutamate metabolites and lower levels of glutamine, arginine, proline and glucose metabolites in the administered groups. It was shown that (−)-gossypol acetate, (+)-gossypol acetate and the positive control drug had similar effects; all of them could regulate the glycolytic metabolism, amino acid metabolism and energy metabolism disorders caused by uterine leiomyomas in the body, improve the immune function of the body and play a positive role in the treatment and prevention of uterine leiomyomas.

Metabolomics is an emerging technology, but there are still some limitations to the analysis of its data. Metabolomics generates a very large amount of data which can only be analyzed and interpreted through sophisticated statistical methods and pattern recognition techniques. Although there are software and algorithms that can help with this data analysis, they can usually only handle specific types of data and the reliability of their analyses needs to be further verified. Sample preparation is also a critical step in metabolomics research, which directly affects the results of subsequent analyses. However, deviations in the sample preparation process may lead to inaccurate results.

Although modern analytical techniques can detect a large number of metabolites, their identification is still a challenge, especially for those metabolites with a low content or complex structure.

### 3.2. Network Toxicology Prediction of Gossypol Optical Isomers

Our research group has carried out preliminary pharmacodynamic experiments which have proven that both (−)-acetate gossypol and (+)-acetate gossypol have effects on the liver and kidney and that the effect of (+)-acetate gossypol on liver function is more obvious [21].

The potential liver injury targets of gossypol isomers were predicted by network toxicology, and their interaction showed that HSP90AA1, HSP90AB1, SRC, MAPK1, AKT1, EGFR, BCL2 and CASP3 were highly related to liver injuries. Through a further KEGG enrichment analysis, it was found that the molecular mechanism by which gossypol optical isomers induced liver injury in rats may be related to the pathways activated in cancer, the PPAR signaling pathway, glycolysis/glycolysis gluconeogenesis and Th17 cell differentiation.

SRC can phosphorylate STAT3, AKT and EGFR to regulate various biological activities [27,28,29]. SRC is engaged in the activation of HSCs and hepatic fibrosis, while activation of primary hepatic stellate cells (HSCs) and hepatic fibrosis is associated with an increase in SRC family kinases [30]. AKT promotes the proliferation, migration and transcription of cells while disrupting apoptosis [31]. HSCs’ proliferation and migration are important for wound healing and liver fibrosis during liver injury [32]. Studies have shown that acetaldehyde and lipopolysaccharide lead to a remarkable increase in HSC proliferation and migration. Silencing Akt1 and Akt2 reduces acetaldehyde- and lipopolysaccharide-mediated proliferation [31]. AKT1 is a protein upstream of the PI3K/Akt signaling channel. AKT1 phosphorylation activates this pathway, thereby affecting tumor cell multiplication and apoptosis [33]. Bcl2 is widely recognized as an important anti-apoptotic molecule in both tumor and normal cells [34,35]. CASP3 is an effector caspase that causes the fragmentation of nuclear DNA in cells during apoptosis [36]. CASP3 is also involved in the MAPK signaling pathway and inflammatory responses [37]. The hyperactivation of CASP3 is strongly associated with myocardial infarction, alcoholic hepatitis, hepatitis B and other diseases [38]. Changes in the PPAR signaling pathway may induce hepatic lipid metabolism disorders [39]. Increased PPAR protein expression is a common feature of steatotic livers. PPAR contributes to the maintenance of the steatosis phenotype of hepatocytes [40]. PPARγ can activate the expression of genes related to TG accumulation in hepatocytes and promote the production of a fatty liver [41]. An important indicator of the progression of liver fibrosis is an abnormal expression of Th17 cells and associated cytokines [42]. Large numbers of Th17 cells have been reported in patients with hepatitis B and cirrhosis, and experiments have shown that an excess of Th17 cells can advance liver fibrosis [43]. Other studies in mice have shown that liver fibrosis is associated with an abnormal increase in Th17 cells and a high expression of Th17-related cytokines [44,45]. The liver is a complex and critical organ for glucose, fatty acid and amino acid metabolism, which has a broad impact on systemic metabolism [46]. In the early stages of liver injury or inflammation, the hepatocyte microenvironment becomes hypoxic, resulting in a failure of oxidative energy production and a switch to the glycolytic and gluconeogenic pathways of producing ATP [47].

Network toxicology is an emerging discipline that combines network science and toxicology, using network analysis techniques to predict and assess the toxicity of chemical substances. It has unique advantages, the first of which is rapid screening: network toxicology is able to quickly identify and assess potentially toxic substances through database searching and network analyses, thus shortening research time. The second is its high-throughput analyses: by using high-throughput sequencing technology, network toxicology can analyze the expression of thousands of genes at once to gain a more comprehensive understanding of the effects of a toxicant on a cell or organism. However, there are data-dependent drawbacks; network toxicology relies on a large number of data resources, and this may lead to inaccurate predictions if the data are incomplete or contain errors. Network toxicology is based on virtual computing and database searching, and further in vivo and in vitro experiments are required if the mechanism of liver injury of specific spin isomers from medroxyprogesterone acetate is to be explored. Future research from our group will be centered around particular pathways and use cell and animal experiments to further explore the mechanism of the hepatotoxicity of GA.

### 3.3. The Significance of This Study

Spin splitting is important in clinical medicine and drug development for several reasons: (1) It improves drug safety. Many drug molecules are chiral, i.e., they exist in both left- and right-handed forms, and the metabolism, efficacy and toxicity of these two forms of drugs in the human body may be different. Through spin splitting, the left-rotation and right-rotation forms of these drugs can be separated and studied separately, allowing us to better understand the mechanism of the drug’s action and improve drug safety. (2) It can optimize the efficacy of drugs. Spin splitting can help us understand the absorption, distribution, metabolism and excretion processes of different chiral forms of drugs in the body. For example, the left-handed form of some drugs may have better efficacy, while their right-handed form may have no efficacy or poor efficacy. Through spin splitting, the most effective form of a drug can be identified and the efficacy of that drug can be improved. (3) It can lead to a reduction in drug side effects. Different chiral forms of a drug may have different metabolic and excretory processes in the body, which may lead to different side effects. Through spin splitting, the left-handed and right-handed forms of a drug can be studied separately, thus allowing us to reduce the side effects of a drug.

Our group found that both (−)-gossypol acetate and (+)-gossypol acetate caused hypokalemic reactions by determining the concentration of potassium ions in rat serum. By determining the ALT, ALP, AST, CRE and BUN in the serum, it was found that the levels of ALT, ALP, AST and BUN were significantly elevated in all dosing groups compared to the normal group, while the levels of CRE were also elevated to varying degrees. Moreover, only ALP was significantly lower in the (−)-gossypol acetate group than in the (+)-gossypol acetate group. This suggests that both (−)-gossypol acetate and (+)-gossypol acetate have an effect on the liver and kidney and that (+)-gossypol acetate has a more pronounced effect on liver function. The mechanism of action of the hepatic injury caused by gossypol acetate needs to be followed up on through detailed experiments.

## 4. Materials and Methods

### 4.1. Drugs and Reagents

The following compounds were used in this study: D_2_O (American CIL Corporation, Room 1106, North 3rd Floor, No. 58, Rangchun Road, Vertical New Town, Chongming District, Shanghai, China), dipotassium hydrogen phosphate (Tianjin Guangfu Fine Chemical Co., Ltd., Nankai University Farm, Nankai District, Tianjin, China), sodium dihydrogen phosphate (Tianjin Guangfu Fine Chemical Research Institute), and sodium chloride (Tianjin GuangFU Technology Development Co., Ltd., No.29 Huacheng Middle Road, Caozili Township, Wuqing District, Tianjin, China).

### 4.2. Laboratory Animals

One hundred and seventeen healthy, clean-grade, 8-week-old, sexually mature SD rats that were female, not pregnant and had a body mass of (180 ± 20) g were selected for this study and provided by the Animal Experimentation Centre of Xinjiang Medical University, License No.: SCXK (Xin) 2018-0003. Due to the effect of the rat sexual cycle on their estrogen and progesterone levels, rats born on the same week were selected for modeling. The animals were housed in the Animal Experimentation Centre of Xinjiang Medical University in an animal laboratory that was an SPF environment with a room temperature of 20 ± 2 °C and a relative humidity of 20–40%. All the rats were given water and food and acclimatized for 1 week. All the experimental procedures were approved by the Animal Ethics Committee of Xinjiang Medical University.

### 4.3. Animal Grouping, Modeling, and Drug Administration

These SD rats were randomly divided into the following nine groups: the normal control group; model control group; positive control group (administered compound medroxyprogesterone acetate tablets); high-, medium- and low-dose (−)-gossypol acetate groups; and high-, medium- and low-dose (+)-gossypol acetate groups. The normal control group was injected intraperitoneally with saline, at 1 mL/100 g, once daily for 6 weeks, while the remaining eight groups were injected intraperitoneally with 0.5 mg/kg of estradiol benzoate once daily and received an intramuscular progesterone injection of 4 mg/kg once every weekday for 5 weeks; this was changed to a simultaneous injection of both hormones at the same dosage in the 6th week. At the end of the modeling period, one rat was randomly selected from each group to observe the formation of uterine fibroids in each group, and their uterus-related indexes were examined to determine whether each model was successfully established.

After the end of modeling, the rats in each group were subjected to a drug intervention: the normal and model control groups were gavaged daily with equal volumes of saline, while the positive control group was administered a dose of 20 mg/kg (equivalent to 20 times the daily dose–weight ratio of adults). The suspension was made with drinking water as the solvent and administered once a day; the volume of the gavage was l mL/100 g. The drugs administered to the (–)-gossypol acetate and (+)-gossypol acetate groups were compound preparations, the main drug of which was (–)-gossypol acetate or (+)-gossypol acetate, while their excipients were vitamin B_1_, B_6_ and potassium chloride (with the drug-positive compound gossypol acetate tablets as a reference: gossypol acetate, 20 mg; vitamin B_1_, 10 mg; vitamin B_6_, 10 mg; potassium chloride, 250 mg). The dose of the main drug was the standard adhered to; the concentrations of the low-, medium- and high-dose groups were 25 mg/kg, 50 mg/kg and 100 mg/kg. We added the corresponding dose of the excipients, which are documented in Table 1, made a suspension with drinking water as the solvent, and administered this once a day at a gavage volume of l mL/100 g for 4 weeks (Table 3).

### 4.4. Sample Collection

After the completion of the gavage, the rats were anesthetized (0.35 mL/100 g) with 10% chloral hydrate solution administered via an intraperitoneal injection. After anesthesia, abdominal aorta blood was taken from each group, and after the blood was placed at room temperature for more than 6 h, its serum was centrifuged at 3500 r/min for 10 min and then isolated and stored in a −80 °C refrigerator for later use.

### 4.5. Untargeted Serum Metabolomics Studies

#### 4.5.1. Sample Preparation

The configuration of the phosphate-buffered solution used was as follows: 10 mL of D_2_O and 40 mL of ultrapure water were used as the solutions, and K_2_HPO_4_ at 0.4169 g, NaH_2_PO_4_ at 0.0713 g and NaCl at 0.4519 g were placed in these solutions and mixed by shaking at pH = 7.0. The rat serum sample was taken out of the −80 °C freezer and thawed at 4 °C, and then 200 μL of it was accurately pipetted, along with 400 μL of phosphate-buffered solution, into a centrifuge tube. The sample was allowed to stand for 10 min at room temperature and then centrifuged at 10,000 r/min and 4 °C for 10 min, before 550 μL of the upper part of the clarified liquid was pipetted into a 5 mm NMR tube, and the processed sample was stored in a 4 °C freezer for further measurements.

#### 4.5.2. 1H-NMR Test of Serum

Serum samples were measured using a 600 M NMR spectrometer, using the Cpmg pulse train mode. The test temperature was 30 °C, the cumulative number of scans was 128, the sampling data point was 32 k, the spectral width was 104 Hz and the water peak was suppressed using the presaturation method. The 1H-NMR spectra of the serum samples from each group were recorded using the same NMR spectrometer, which was conducive to the identification of their metabolic components.

#### 4.5.3. 1H-NMR Pattern Processing

The 1H-NMR NMR spectra of each group of rat serum were processed and analyzed using NMR processing software (MestReNova15). The lactic acid chemical shift value (δ1.331 ppm) was used as the standard calibration for manual correction, the baseline was manually corrected, the δ4.68~δ5.10 ppm water peak region was removed, the spectra of the δ0.10~δ9.00 ppm region were segmented into equal widths, and all the maps were segmented with an integration interval of 0.003 ppm. The obtained integral data were normalized for multivariate statistical analyses.

#### 4.5.4. Statistical Processing

SIMCA software14.1 was used to perform the partial least squares discriminant analysis (PLS-DA), orthogonal partial least squares discriminant analysis (OPLS-DA) and displacement test, and R^2^X and Q^2^ were used as the quality evaluation indicators of the established models. R^2^X describes the optimization degree of the model, R^2^Y describes the percentage of variation in the reaction variable Y, and the cumulative prediction degree of the model is described by the cross-check parameter Q^2^, which indicates the authenticity of the prediction results. In this experiment, the metabolite correlation coefficient was used to determine whether the metabolites were different between the groups, and α = 0.05 was used as the test standard. Significant differences in Pearson’s correlation coefficient |r| > 0.632 (n = 8) were used to detect whether the change in the metabolite content had a significance threshold. Metabolites represented by correlation coefficients |r| > 0.632 are statistically significant. Larger values of |r| indicate greater variability.

### 4.6. Network Toxicology Studies

#### 4.6.1. Obtaining Information about the Gossypol Acetate Compound

We download the sdf. two-dimensional conformational format map of gossypol acetate and the canonical SMILE sequence from the official website of Pubchem for our network toxicological analysis.

#### 4.6.2. Drug Target Acquisition

We uploaded the sdf. format to the PharmMapper platform, set the “reserved target match number” to 300, obtained the drug target of gossypol acetate and then imported the protein target UniProt ID number of gossypol acetate in the UniProt KB search interface of the UniProt database. We then selected “Homo sapiens” and obtained the gene targets of gossypol acetate after its retrieval and transformation. The SMILE sequence file of gossypol acetate was imported into the SwissTargetPrediction platform, and the species was also set to “Homo sapiens” to obtain the potential gene targets of gossypol acetate. We finally integrated the target components predicted by the PharmMapper and Swiss Target Prediction technology platforms and removed duplicates to obtain the final target components of gossypol acetate.

#### 4.6.3. Hepatotoxicity Target Acquisition

By entering the keywords “liver toxicity, liver damage, liver disease, liver harm” into the GeneCards database, the Comparative Toxicogenomics Database (CTD) and Online Mendelian Inheritance in Man (OMIM), the reported gene targets related to liver damage were searched, duplicate genes and false-positive genes were removed and disease targets related to liver toxicity were obtained.

#### 4.6.4. Common Target Acquisition

The gossypol acetate target components obtained in Section 4.6.2 and the hepatotoxicity-related targets obtained in Section 4.6.3 were introduced into the Vennn2.1.0 platform https://bioinfogp.cnb.csic.es/tools/venny/ (accessed on 2 August 2023) for the screening of common targets.

#### 4.6.5. Protein Interaction Network Construction and Analysis

The common targets obtained in Section 4.6.4 were imported into the STRING database, the species was limited to humans, their protein–protein interactions were obtained, the results were imported into Cytoscape software3.9.1 to map the PPI network and the PPI network was then analyzed. The size and color of the nodes in the PPI network diagram are related to the “degree” of the node; that is, the larger the “degree” value of the node is, the larger the node is and the redder its color is. The width of an edge indicates the strength of the interaction between the two nodes connected by that edge; that is, the stronger the interaction, the wider the edge.

#### 4.6.6. GO Bioprocess and KEGG Pathway Enrichment Analysis

We imported the common targets in Section 4.6.4 into the Metascape database, set the species to “Homo sapiens”, ran GO biological processes and KEGG pathway enrichment analyses and obtained GO analysis results that included biological processes (BPs), molecular functions (MFs) and cellular components (CCs). The results of the KEGG enrichment analysis were saved as TSV format files. The results were imported into the bioinformatics http://www.bioinformatics.com.cn/ (accessed on 2 August 2023) platform for visualization and mapping, and a signal pathway bubble map was drawn.

## 5. Conclusions

In our metabolomics experiment, the levels of isoleucine, alanine, lysine, glutamic acid, glycine, proline, acetoacetate, choline, pyruvate, lactic acid, γ-aminobutyric acid, lipids (including LDL and VLDL), cholesterol, creatine, unsaturated fatty acids and other metabolites in the serum from the model group were significantly decreased. The levels of glucose, glycerol, arginine and citrulline were increased. These metabolites together constitute the metabolomic characteristics of the uterine fibroid model rat. Uterine fibroids caused a disorder of amino acid metabolism in the body and led to abnormal energy metabolism, weakened immune function and abnormal glycolysis and gluconeogenesis metabolism. The effects of (−)-gossypol acetate and (+)-gossypol acetate are similar to those of the positive control drug, which regulates the glycolysis and gluconeogenesis metabolism, amino acid metabolism and energy metabolism disorders caused by uterine fibroids, improves the body’s immune ability and plays a positive role in the treatment and prevention of uterine fibroids.

Our research group’s pharmacodynamic experiments have proven that the therapeutic effects of gossypol optical isomers [(−)-acetate gossypol and (+)-acetate gossypol] in the treatment of uterine fibroids are different. We also discussed some of the possible toxic reactions caused by the two drugs. The mechanism of liver injury induced by gossypol optical isomers in the treatment of uterine fibroids was elucidated via network toxicology. This has important implications for the development of single optical isomer drugs for the treatment of uterine fibroids.

## Figures and Tables

**Figure 1 pharmaceuticals-17-01363-f001:**
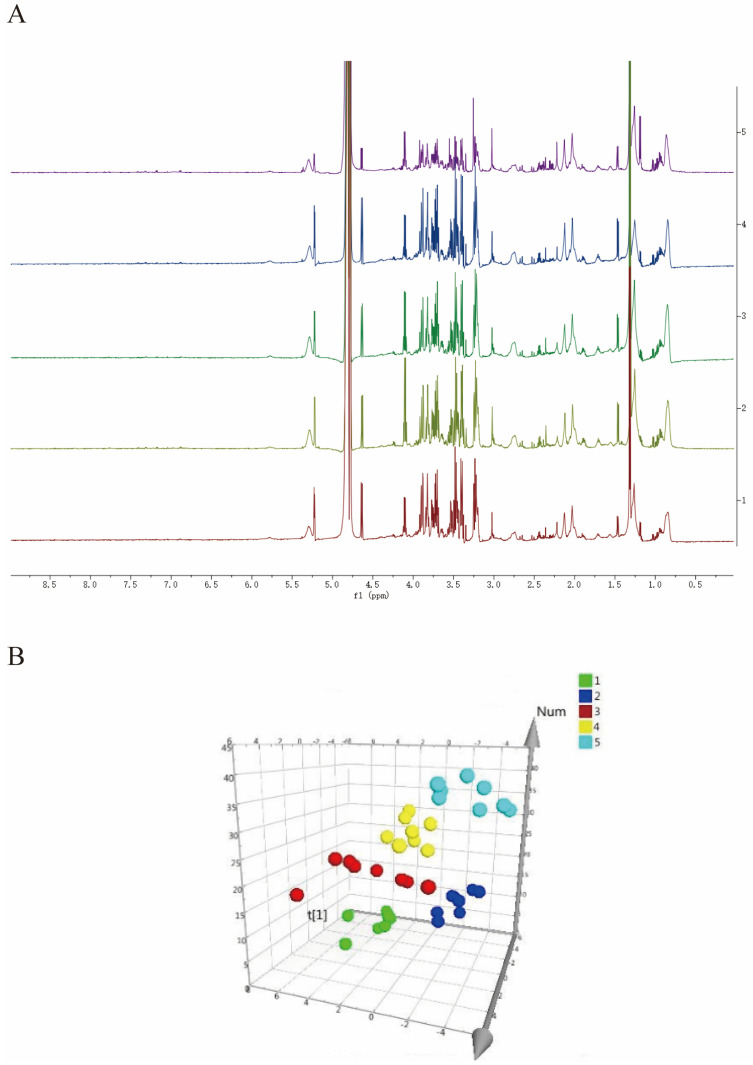
(**A**) 1H-NMR differential metabolite spectra of the serum from rats in each group. (1) Blank group; (2) uterine fibroid model group; (3) high-dose (+)-gossypol acetate group; (4) high-dose (−)-gossypol acetate group; and (5) positive control group. (**B**) Three-dimensional spatial distribution map of PLS-DA analysis. (1: positive control group; 2: high-dose (+)-gossypol acetate group; 3: high-dose (−)-gossypol acetate group; 4: model control group 5: normal control group). The corresponding metabolites 1–27 are listed in detail in Table 1.

**Figure 2 pharmaceuticals-17-01363-f002:**
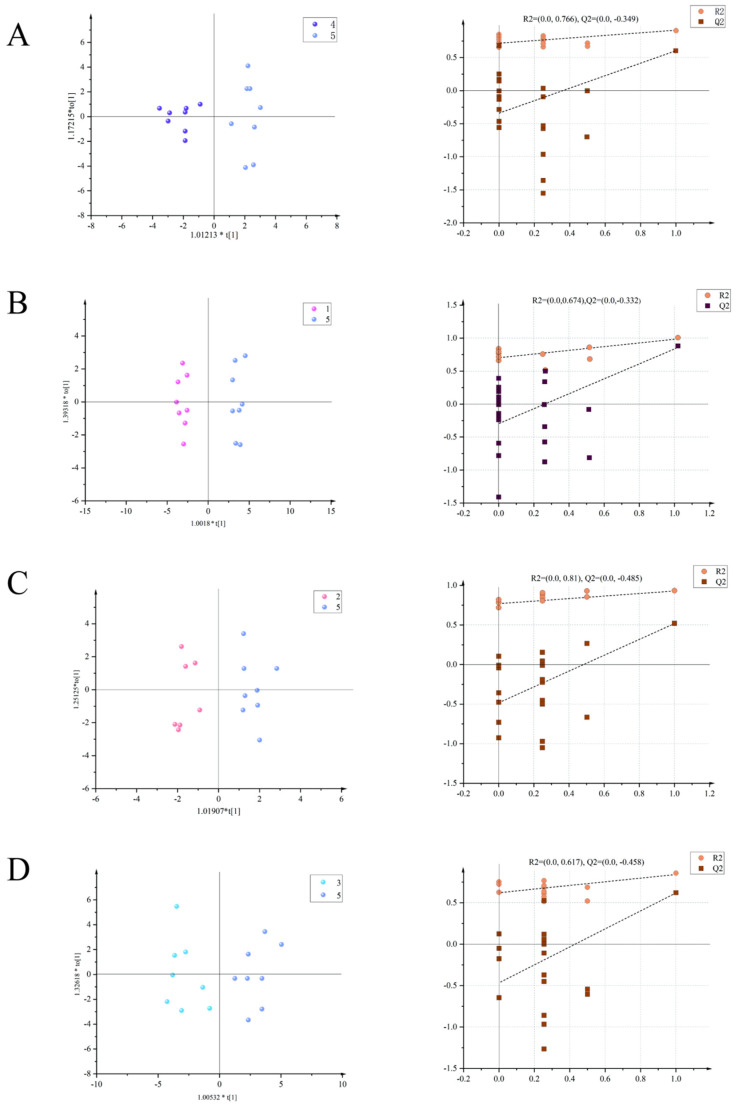
Serum OPLS-DA scores of the rats in each group (**left**) with model verification (**right**). (**A**) Positive control group vs. model group; (**B**) high-dose (+)-gossypol acetate group vs. model group; (**C**) high-dose (−)-gossypol acetate group vs. model group; and (**D**) normal control group vs. model group.

**Figure 3 pharmaceuticals-17-01363-f003:**
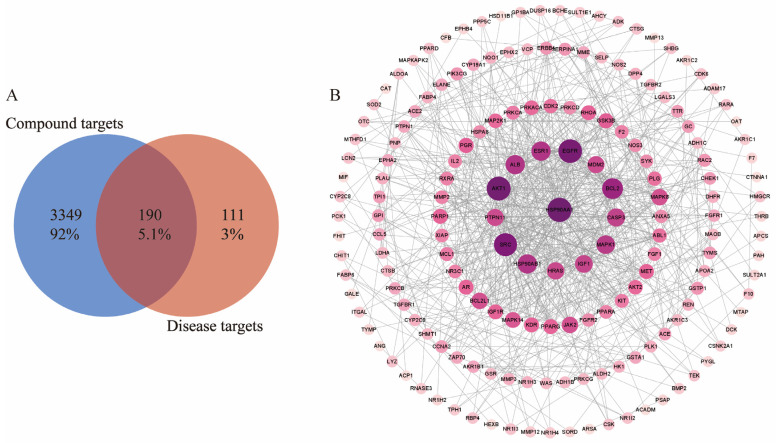
(**A**) Common targets of gossypol isomer-induced hepatotoxicity. (**B**) “component-target interaction” PPI network diagram (**C**) GO enrichment analysis of targets related to gossypol isomers. (**D**) KEGG enrichment analysis of targets related to gossypol isomers.

**Table 2 pharmaceuticals-17-01363-t002:** Main differential metabolites in serum of rats in each group.

Serial Number	Metabolites	Chemical Shift	Ascription
1	Cholesterol	0.84 (m)	C26, C27
2	Isoleucine	0.94 (t)	δ-CH_3_
3	Leucine	0.98 (d)	δ-CH_3_
4	Lipid	1.26 (m), 1.57 (m), 2.23 (m)	CH_3_CH_2_(CH_2_)_n_, CH_2_CH_2_CO, CH_2_CO
5	Lactic acid	1.33 (d), 4.10 (q)	CH_3_, CH
6	Alanine	1.46 (d)	CH_3_
7	Acetic acid	1.91 (s)	CH_3_
8	Lysine	1.91 (m)	β-CH_2_
9	Glutamic acid	2.00 (m), 2.12 (m)	half β-CH_2_
10	Methionine	2.12 (s)	S-CH_3_
11	Acetoacetate	2.22 (s)	CH_3_
12	Glutamine	2.42 (m)	half γ-CH_2_
13	Pyruvic acid	2.35 (s)	CH_3_
14	Citrate	2.52 (d)	half CH_2_
15	γ-aminobutyric acid	3.02 (t)	γ-CH_2_
16	Choline	3.66 (m), 4.29 (m)	NCH_2_, OCH_2_
17	Arginine	3.24 (t)	δ-CH_2_
18	Proline	3.45 (m)	half δ-CH_2_
19	β-glucose	3.48 (t), 4.64 (d)	H3, H1
20	α-glucose	3.53 (dd), 3.82 (m), 3.76 (dd), 5.22 (d)	H2, half CH_2_-C6, H1
21	Glycine	3.54 (s)	CH_3_
22	Citrulline	3.71 (m)	δ-CH
23	Glycerin	3.89 (m)	C2-H
24	Creatine	3.92 (s)	CH_2_
25	Unsaturated fatty acids	5.28 (m)	CH=CHCH_2_CH=CH
26	Urea	5.77 (s)	NH_2_
27	Formic acid	8.44 (s)	CH

Note: s, single peak; d, double peak; t, triple peak; m, multiple peaks; dd, doublet of doublets.

**Table 3 pharmaceuticals-17-01363-t003:** Dosage rationing table of (−)-gossypol acetate and (+)-gossypol acetate combination drugs (mg).

Groups	(–)-GA	(+)-GA	VB_1_	VB_6_	KCL
(–)-GA-L	25	—	12	12	312
(–)-GA-M	50	—	25	25	625
(–)-GA-H	100	—	50	50	1250
(+)-GA-L	—	25	12	12	312
(+)-GA-M	—	50	25	25	625
(+)-GA-H	—	100	50	50	1250

## Data Availability

Data is contained within the article.

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
