# Peer review of "Metabolomic Profiling and Network Toxicology: Mechanistic Insights into Effect of Gossypol Acetate Isomers in Uterine Fibroids and Liver Injury"

_pharmaceuticals, 2024, doi:10.3390/ph17101363_

Round 1

Reviewer 1 Report

Comments and Suggestions for Authors

Manuscript title "Nontargeted metabolomics and network toxicology to investigate the mechanism of action of gossypol optical isomer on uterine fibroids and liver injury."
The manuscript presents an interesting investigation into the effects of gossypol optical isomers on uterine fibroids and liver injury using serum metabolomics and network toxicology.
Overall, the study is well-designed, and the results provide valuable insights. However, there are a few suggestions that would help strengthen the manuscript before it can be accepted for publication.
1. Clarification of experimental design: Please provide more details on the distribution of the 120 female rats across the different groups (control, uterine fibroid model, positive control, and the various gossypol acetate treatment groups).
2. Clearly state the dosages used for the (+)-gossypol acetate and (-)-gossypol acetate groups.
3. Metabolomics data analysis: Provide more information on the statistical methods used for the metabolomics data analysis, including the software and algorithms used for differential metabolite identification and pathway analysis.
4. Discuss the limitations of the metabolomics approach and how it may have influenced the identified metabolites.
5. Network toxicology: Expand the description of the network toxicology methods, including the databases, algorithms, and software used for target and pathway identification.
6. Discuss the strengths and limitations of the network toxicology approach in the context of this study. Discussion: Provide a more comprehensive discussion on the potential mechanisms by which the gossypol optical isomers exert their effects on uterine fibroids and liver injury, drawing connections between the metabolomics and network toxicology findings.
7. Discuss the clinical relevance of the findings and the potential implications for the treatment of uterine fibroids and liver injury.
8. Minor formatting and language: Ensure consistent use of abbreviations throughout the manuscript. Check for any grammatical or typographical errors.

Comments on the Quality of English Language

Ensure consistent use of abbreviations throughout the manuscript. Check for any grammatical or typographical errors.

Author Response

Response letter

Dear editors and reviewers,

We are grateful for your constructive comments and suggestions for our manuscript entitled “Using metabolomics to investigate the mechanism of action of optical isomers of gossypol acetate on uterine fibroids and predict the potential mechanism of the liver injury they cause using network toxicology” (ID: pharmaceuticals-3144089). Your comments are very valuable and helpful for improving our manuscript. In the following, the responses to all the comments are provided one by one.

We have tried our best to make all the revisions clear, and we hope that the revised manuscript can satisfy the requirements for publication.

The main revisions in the new manuscript are:

Reviewer 1

  1. Clarification of experimental design: Please provide more details on the distribution of the 120 female rats across the different groups (control, uterine fibroid model, positive control, and the various gossypol acetate treatment groups).

Reply:

One hundred and seventeen healthy, clean-grade, 8-week-old, sexually mature SD rats that were female, not pregnant and had a body mass of (180±20) g were selected for this study and provided by the Animal Experimentation Centre of Xinjiang Medical University, License No.: SCXK (Xin) 2018-0003. Due to the effect of the rat sexual cycle on their estrogen and progesterone levels, rats born on the same week were selected for modeling. The animals were housed in the Animal Experimentation Centre of Xinjiang Medical University in an animal laboratory that was an SPF environment with a room temperature of 20±2°C and a relative humidity of 20%-40%. All the rats were given water and food and acclimatized for 1 week. All the experimental procedures were approved by the Animal Ethics Committee of Xinjiang Medical University.

These SD rats were randomly divided into the following nine groups: the normal control group; model control group; positive control group (administered compound medroxyprogesterone acetate tablets); high-, medium- and low-dose (-) -gossypol acetate groups; and high-, medium- and low-dose (+)-gossypol acetate groups. The normal control group was injected intraperitoneally with saline, at 1 mL/100 g, once daily for 6 weeks, while the remaining eight groups were injected intraperitoneally with 0.5 mg/kg of estradiol benzoate once daily and received an intramuscular progesterone injection of 4 mg/kg once every weekday for 5 weeks; this was changed to a simultaneous injection of both hormones at the same dosage in the 6th week. At the end of the modeling period, one rat was randomly selected from each group to observe the formation of uterine fibroids in each group, and their uterus-related indexes were examined to determine whether each model was successfully established.

  1. Clearly state the dosages used for the (+)-gossypol acetate and (-)-gossypol acetate groups.

Reply:

Dosage rationing table of (-)-gossypol acetate and (+)-gossypol acetate combination drugs (mg).

Groups

(–)-GA

(+)-GA

VB1

VB6

KCL

(–)-GA-L

25

12

12

312

(–)-GA-M

50

25

25

625

(–)-GA-H

100

50

50

1 250

(+)-GA-L

25

12

12

312

(+)-GA-M

50

25

25

625

(+)-GA-H

100

50

50

1 250

  1. Metabolomics data analysis: Provide more information on the statistical methods used for the metabolomics data analysis, including the software and algorithms used for differential metabolite identification and pathway analysis.

Reply:

In this experiment, the metabolite correlation coefficient was used to determine whether the metabolites were different between the groups, and α=0.05 was used as the test standard. Significant differences in Pearson's correlation coefficient |r|>0.632 (n=8) were used to detect whether the change in the metabolite content had a significance threshold. Metabolites represented by correlation coefficients |r|>0.632 are statistically significant. Larger values of |r| indicate greater variability.

  1. Discuss the limitations of the metabolomics approach and how it may have influenced the identified metabolites.

Reply:

Metabolomics is an emerging technology, but there are still some limitations to the analysis of its data. Metabolomics generates a very large amount of data which can only be analyzed and interpreted through sophisticated statistical methods and pattern recognition techniques. Although there are software and algorithms that can help with this data analysis, they can usually only handle specific types of data and the reliability of their analyses needs to be further verified. Sample preparation is also a critical step in metabolomics research, which directly affects the results of subsequent analyses. However, deviations in the sample preparation process may lead to inaccurate results.

Although modern analytical techniques can detect a large number of metabolites, their identification is still a challenge, especially for those metabolites with a low content or complex structure.

  1. Network toxicology: Expand the description of the network toxicology methods, including the databases, algorithms, and software used for target and pathway identification.

Reply:

4.6 Network toxicology studies

4.6.1 Obtaining information about the gossypol acetate compound

We download the sdf. two-dimensional conformational format map of gossypol acetate and the canonical SMILEs sequence from the official website of Pubchem for our network toxicological analysis.

4.6.2 Drug target acquisition

We uploaded the sdf. format to the PharmMapper platform, set the "reserved target match number" to 300, obtained the drug target of gossypol acetate and then imported the protein target UniProt ID number of gossypol acetate in the UniProt KB search interface of the UniProt database. We then selected "Homo sapiens" and obtained the gene targets of gossypol acetate after its retrieval and transformation. The SMILEs sequence file of gossypol acetate was imported into the SwissTargetPrediction platform, and the species was also set to "Homo sapiens" to obtain the potential gene targets of gossypol acetate. We finally integrated the target components predicted by the PharmMapper and Swiss Target Prediction technology platforms and removed duplicates to obtain the final target components of gossypol acetate.

4.6.3 Hepatotoxicity target acquisition

By entering the keywords "liver toxicity, liver damage, liver disease, liver harm" into the GeneCards database, the Comparative Toxicogenomics Database (CTD) and Online Mendelian Inheritance in Man (OMIM), the reported gene targets related to liver damage were searched, duplicate genes and false positive genes were removed and disease targets related to liver toxicity were obtained.

4.6.4 Common target acquisition

The gossypol acetate target components obtained in Section "4.6.2" and the hepatotoxicity-related targets obtained in Section "4.6.3" were introduced into the Vennn2.1.0 platform (https://bioinfogp.cnb.csic.es/tools/venny/) for the screening of common targets.

4.6.5 Protein interaction network construction and analysis

The common targets obtained in "4.6.4" were imported into the STRING database, the species was limited to human, their protein–protein interactions were obtained, the results were imported into Cytoscape software to map the PPI network and the PPI network was then analyzed. The size and color of the nodes in the PPI network diagram are related to the "degree" of the node; that is, the larger the "degree" value of the node is, the larger the node is and the redder its color is. The width of an edge indicates the strength of the interaction between the two nodes connected by that edge; that is, the stronger the interaction, the wider the edge.

4.6.6 GO bioprocess and KEGG pathway enrichment analysis

We imported the common targets in Section "4.6.4" into the Metascape database, set the species to "Homo sapiens", ran GO biological processes and KEGG pathway enrichment analyses and obtained GO analysis results that included biological processes (BPs), molecular functions (MFs) and cellular components (CCs). The results of the KEGG enrichment analysis were saved as TSV format files. The results were imported into the bioinformatics (http://www.bioinformatics.com.cn/) platform for visualization and mapping, and a signal pathway bubble map was drawn.

  1. Discuss the strengths and limitations of the network toxicology approach in the context of this study. Discussion: Provide a more comprehensive discussion on the potential mechanisms by which the gossypol optical isomers exert their effects on uterine fibroids and liver injury, drawing connections between the metabolomics and network toxicology findings.

Reply:

Network toxicology is an emerging discipline that combines network science and toxicology, using network analysis techniques to predict and assess the toxicity of chemical substances. It has unique advantages, the first of which is rapid screening: network toxicology is able to quickly identify and assess potentially toxic substances through database searching and network analyses, thus shortening research time. The second is its high-throughput analyses: by using high-throughput sequencing technology, network toxicology can analyze the expression of thousands of genes at once to gain a more comprehensive understanding of the effects of a toxicant on a cell or organism. However, there are data-dependent drawbacks; network toxicology relies on a large number of data resources, and this may lead to inaccurate predictions if the data are incomplete or contain errors. Network toxicology is based on virtual computing and database searching, and further in vivo and in vitro experiments are required if the mechanism of liver injury of specific spin isomers from medroxyprogesterone acetate is to be explored. Future research from our group will be centered around particular pathways and use cell and animal experiments to further explore the mechanism of the hepatotoxicity of GA.

  1. Discuss the clinical relevance of the findings and the potential implications for the treatment of uterine fibroids and liver injury.

Reply:

Spin splitting is important in clinical medicine and drug development for several reasons: (1) It improves drug safety. Many drug molecules are chiral; i.e., they exist in both left- and right-handed forms, and the metabolism, efficacy and toxicity of these two forms of drugs in the human body may be different. Through spin splitting, the left-rotation and right-rotation forms of these drugs can be separated and studied separately, allowing us to better understand the mechanism of the drug’s action and improve drug safety. (2) It can optimize the efficacy of drugs. Spin splitting can help us understand the absorption, distribution, metabolism and excretion processes of different chiral forms of drugs in the body. For example, the left-handed form of some drugs may have better efficacy, while their right-handed form may have no efficacy or poor efficacy. Through spin splitting, the most effective form of a drug can be identified and the efficacy of that drug can be improved. (3) It can lead to a reduction in drug side effects. Different chiral forms of a drug may have different metabolic and excretory processes in the body, which may lead to different side effects. Through spin splitting, the left-handed and right-handed forms of a drug can be studied separately, thus allowing us to reduce the side effects of a drug.

Our group found that both (-)-gossypol acetate and (+)-gossypol acetate caused hypokalemic reactions by determining the concentration of potassium ions in rat serum. By determining the ALT, ALP, AST, CRE and BUN in the serum, it was found that the levels of ALT, ALP, AST and BUN were significantly elevated in all dosing groups compared to the normal group, while the levels of CRE were also elevated to varying degrees. Moreover, only ALP was significantly lower in the (-)-gossypol acetate group than in the (+)-gossypol acetate group. This suggests that both (-)-gossypol acetate and (+)-gossypol acetate have an effect on the liver and kidney and that (+)-gossypol acetate has a more pronounced effect on liver function. The mechanism of action of the hepatic injury caused by gossypol acetate needs to be followed up on through detailed experiments.

  1. Minor formatting and language: Ensure consistent use of abbreviations throughout the manuscript. Check for any grammatical or typographical errors.

Reply:We have an English editing service

Reviewer 2 Report

Comments and Suggestions for Authors

Dear Authors,

Below you can find some of my suggestions concerning your manuscipt

- L11 Why have you started with authors contribution when there is a dedicated section about it

-Please, consider revision your abstract as now it sound rather scholar than research

- Revise references according to journals style in text

- Unfortunately, fig. 2 at its current state is unreadable

- To me, it is hard to follow the discussion in your work

Comments on the Quality of English Language

I would suggest that you revise your manuscript to make it sound more natural. For example, the verb used in L73 is rather inappropriate.

Author Response

Response letter

Dear editors and reviewers,

We are grateful for your constructive comments and suggestions for our manuscript entitled “Using metabolomics to investigate the mechanism of action of optical isomers of gossypol acetate on uterine fibroids and predict the potential mechanism of the liver injury they cause using network toxicology” (ID: pharmaceuticals-3144089). Your comments are very valuable and helpful for improving our manuscript. In the following, the responses to all the comments are provided one by one.

We have tried our best to make all the revisions clear, and we hope that the revised manuscript can satisfy the requirements for publication.

The main revisions in the new manuscript are:

Reviewer 2

  1. Why have you started with authors contribution when there is a dedicated section about it

Reply: Dear reviewer, we have now revised it.

  1. Please, consider revision your abstract as now it sound rather scholar than research.

Reply: Dear reviewer, we have revised the abstract in accordance with your comments.

Abstract: Gossypol is a natural polyphenolic dialdehyde product that is primarily isolated from cottonseed. It is a racemized mixture of (-) -gossypol and (+) -gossypol that has anti-infection, antimalarial, antiviral, antifertility, antitumor and antioxidant activities, among others. Gossypol optical isomers have been reported to differ in their biological activities and toxic effects. In this study, we performed a metabolomics analysis of rat serum using 1H-NMR technology to investigate gossypol optical isomers’ mechanism of action on uterine fibroids. Network toxicology was used to explore the mechanism of the liver injury caused by gossypol optical isomers. SD rats were randomly divided into a normal control group; model control group; a drug-positive group (compound gossypol acetate tablets); high-, medium- and low-dose (-) -gossypol acetate groups; and high-, medium- and low-dose (+) -gossypol acetate groups.

Serum metabolomics showed that gossypol optical isomers’ pharmacodynamic effect on rats’ uterine fibroids affected their lactic acid, cholesterol, leucine, alanine, glutamate, glutamine, arginine, proline, glucose, etc. According to network toxicology, the targets of the liver injury caused by gossypol optical isomers included HSP90AA1, SRC, MAPK1, AKT1, EGFR, BCL2, CASP3, etc. KEGG enrichment showed that the toxicity mechanism may be related to pathways active in cancer, such as the PPAR signaling pathway, glycolysis/glycolysis gluconeogenesis, Th17 cell differentiation, and 91 other closely related signaling pathways. (-) -gossypol acetate and (+) -gossypol acetate play a positive role in the treatment and prevention of uterine fibroids. Gossypol optical isomers cause liver damage through multiple targets and pathways.

  1. Revise references according to journals style in text

Reply: Dear reviewer, according to your modification suggestion, we have changed the document format of MDPI with endnote.

4. Unfortunately, fig. 2 at its current state is unreadable

Reply: Dear reviewer, I'm very sorry that you can't see Figure 2, now Figure 2 is put in the following.

5.To me, it is hard to follow the discussion in your work

Reply: Dear reviewer, I am very sorry for giving you an unpleasant impression. Now we will discuss and revise it again. And MDPI's English editor was used.

Reviewer 3 Report

Comments and Suggestions for Authors

The authors describe the results obtaned by using a NMR based metaboolic approach for the study of the mode of action of gossypol optical isomer on uterine fibroids and liver injury.

Although the aim of the study is interesting, the experimental part and discussion should be improved.

-in the section 2.1.1 is not clear the meaning of the title “analysis of rat groups in each group…..”

In the same section is not clear the differences (if there are) between control group and model group  indicated as separate groups in the lines 73-74.

-the authors describe a “non targeted approach” but they used in gthe manuscript a targeted approach (only supervised methods have been presented). A prior unsupervised PCA analysis is strongly suggested.

- Figure 1: It is not necessary to add five figures, I suggest to simplify the figure overlapping the five spectra. This may also help the reader in the visual inspection of the differences among the conditions.

-many typos are present in the manuscript, please check and correct them.

In the section 2.2 the authors used a network toxicology analysis, please describe better the analysis in the first part of the section.

-            Lines 171-172. I don’t understand the mean of the sentence

Comments on the Quality of English Language

Extensive editing of English language required.

Author Response

Response letter

Dear editors and reviewers,

We are grateful for your constructive comments and suggestions for our manuscript entitled “Using metabolomics to investigate the mechanism of action of optical isomers of gossypol acetate on uterine fibroids and predict the potential mechanism of the liver injury they cause using network toxicology” (ID: pharmaceuticals-3144089). Your comments are very valuable and helpful for improving our manuscript. In the following, the responses to all the comments are provided one by one.

We have tried our best to make all the revisions clear, and we hope that the revised manuscript can satisfy the requirements for publication.

The main revisions in the new manuscript are:

Reviewer 3

  1. in the section 2.1.1 is not clear the meaning of the title “analysis of rat groups in each group…..”

Reply: Dear reviewer, we have now revised the title according to your comments. “Analysis and attribution of rat serum’s 1H-NMR metabolite profiles”

  1. In the same section is not clear the differences (if there are) between control group and model group indicated as separate groups in the lines 73-74.

Reply: There was a small gap between the metabolites in each group.

The 1H-NMR spectra of these five groups’ rat serum were segmented, and the resulting integral values were analyzed via PLS-DA to obtain spatial distribution maps (3D plots) (Figure 3E). In the PLS-DA analysis, R2X=0.452, R2Y=0.331, Q2=0.263 and each group occupies an independent space, indicating that each group’s serum has different metabolic components.

  1. The authors describe a “non targeted approach” but they used in gthe manuscript a targeted approach (only supervised methods have been presented). A prior unsupervised PCA analysis is strongly suggested.

Reply:

Because the results of our unsupervised PCA analysis were not good, probably due to the small differences between groups, supervised OPLS-DA was performed. The analysis figure of our PCA is as follows.

  1. Figure 1: It is not necessary to add five figures, I suggest to simplify the figure overlapping the five spectra. This may also help the reader in the visual inspection of the differences among the conditions.

Reply:

  1. many typos are present in the manuscript, please check and correct them.

Reply:

We have used MDPI's English editing service

  1. In the section 2.2 the authors used a network toxicology analysis, please describe better the analysis in the first part of the section.

Reply: We have revised this section.

2.2 Network toxicology analysis

2.2.1 Prediction of targets

A total of 569 target component of gossypol acetate were predicted by PharmMapper and the SwissTargetPrediction platform. After removing repeated targets, a total of 301 target components were obtained. A total of 190 hepatotoxic targets related to gossypol optical isomers were screened by drawing on the cross-section of a Venn diagram (Figure 3C).

2.2.2 Construction and analysis of protein interaction networks

The 190 targets acquired were imported into the STRING database in order to download the results of their protein interactions; the confidence level was set to 0.700, the discrete nodes were hidden and the results were imported into Cytoscape software to plot their PPI networks (Figure 3D). The degree of inter-target interactions varies, with nodes denoting proteins and edges denoting inter-protein associations. Node sizes and color depths are positively correlated with their degree value. The larger the degree value, the higher the score and the more critical the role of the target. The network graph has 175 nodes and 730 edges; its average node degree is 7.64 and its average local clustering coefficient is 0.433.

2.2.3 GO gene function and KEGG pathway analyses

The results of the GO analysis included three branches: biological processes, molecular functions and cellular components (Figure 3A). There were 474 results for biological processes (BPs), which mainly involved the "response to hormones", "protein phosphorylation", "positive regulation of phosphorylation" and other processes. A total of 28 results were enriched cell components (CCs), and these mainly involved the "vesicle cavity", "receptor complex", "side of membrane", etc. A total of 35 results were obtained from the molecular function (MF) analysis, and these mainly included "phosphotransferase activity, receptor alcohol group", "kinase binding", "oxygen multireductase activity" and other functions.

A total of 91 pathways were enriched in the KEGG pathway analysis. The key pathways included pathways related to cancer, the PPAR signaling pathway, glycolysis/gluconeogenesis and Th17 cell differentiation (Figure 3B).

  1. Lines 171-172. I don’t understand the mean of the sentence.

Reply:

We have used MDPI's English editing service

Round 2

Reviewer 1 Report

Comments and Suggestions for Authors

It is an interesting study, the authors have done metabolomic analysis of rat serum to explore the mechanism of action of gossypol optical isomers on uterine fibroids and explored the liver damage mechanism of gossypol optical isomers using Network toxicology.

There is still room for improvement:

 1. The title needs to be more concise and succinct. One suggestion is as follows: "Metabolomic Profiling and Network Toxicology: Mechanistic Insights into effect of Gossypol Acetate Isomers in Uterine Fibroids and Liver Injury". 

2. The introduction part must focus more on the rational and novelty of this work, what gaps the authors are trying to fill?

3. The figure Quality must be improved, the font size in the figures is small and hardly visible.

Author Response

Response letter

Dear editors and reviewers,

We are grateful for your constructive comments and suggestions for our manuscript entitled “Using metabolomics to investigate the mechanism of action of optical isomers of gossypol acetate on uterine fibroids and predict the potential mechanism of the liver injury they cause using network toxicology” (ID: pharmaceuticals-3144089). Your comments are very valuable and helpful for improving our manuscript. In the following, the responses to all the comments are provided one by one.

We have tried our best to make all the revisions clear, and we hope that the revised manuscript can satisfy the requirements for publication.

The main revisions in the new manuscript are:

Reviewer 1

  1. The title needs to be more concise and succinct. One suggestion is as follows: "Metabolomic Profiling and Network Toxicology: Mechanistic Insights into effect of Gossypol Acetate Isomers in Uterine Fibroids and Liver Injury".

Reply:

After much deliberation, I have finally taken your advice and changed the title to “Metabolomic Profiling and Network Toxicology: Mechanistic Insights into effect of Gossypol Acetate Isomers in Uterine Fibroids and Liver Injury.”

  1. The introduction part must focus more on the rational and novelty of this work, what gaps the authors are trying to fill?

Reply:

Uterine leiomyoma (UL) is a hormone-dependent benign tumor that occurs in the female myometrium and is most common in women aged 30-50 years, with women who are around 50 years old and near menopause representing the peak age of the onset of this disease [1]. Most patients in the early stage of uterine fibroid formation are asymptomatic, as these are often only found incidentally during pelvic or ultrasound examinations. Due to the production of tumor masses in the uterine tissue, the uterine cavity and adjacent organs (the ovaries, fallopian tubes) are compressed, resulting in uterine bleeding, infertility, ectopic pregnancies, spontaneous abortion, anemia and other clinical manifestations [2]. When fibroids grow they mostly protrude from the serosal surface of the uterus, and the blood vessels connected to the tumor mass can easily be twisted and torn, causing severe acute pain in the abdomen [3]. It is generally believed that the abnormal secretion of sex hormones such as estrogen and progesterone is the main reason for the production and growth of uterine fibroids, which can promote mitosis, the proliferation of smooth muscle cells and the growth of leiomyomas [4,5]. Uterine fibroids are benign tumors with a malignancy rate of 0.4% to 1.25% and are usually associated with a risk of fibroid progression when they grow rapidly in the short term [6]. Therefore, reducing fibroid cell proliferation, shrinking the volume of uterine fibroids and removing tumors are the primary therapeutic goals for uterine fibroids.

Uterine fibroids are mainly treated via surgery or the administration of drugs. The former can directly remove the tumor; however, it causes significant trauma and the recurrence rate after surgery is high [2]. In terms of drug treatments, the most commonly used drug is the anti-progestin preparation mifepristone. Another drug compound listed as suitable for treating uterine fibroids is gossypol acetate, which is a non-hormonal drug administered in the form of a tablet, and which can effectively inhibit the secretion of steroidal hormone receptors in the uterine smooth muscle, endometrium and other parts of the uterus. It leads to a reduction or elimination of fibroids and plays a role in treating and relieving symptoms of uterine fibroids. But the elevated hepatic SGPT level and GSH content were induced by mixed gossypol. Different doses of mixed gossypol acetate can cause significant pathological changes in rat liver tissue, such as mitochondrial vacuolization, endoplasmic reticulum dilation, perinuclear space widening and glial fiber proliferation in Disse space. Meanwhile, mixed gossypol acetate produced a large amount of O2 and H2O2, which affected the binding of liver to microsomal proteins[7,8]. It is revealed the effect of mixed gossypol on the liver at the pathological level.

Because of gossypol is a racemate and a natural product of polyphenol bisaldehyde that primarily isolated from cottonseed [9,10]. It has a variety of biological activities, including anti-infective [9], antimalarial [11], antiviral [12], antifertility [13], antitumor [14] and antioxidant activities [15]. The biological activity and toxic effects of (+)-gossypol and (-)-gossypol have been reported to differ [16,17]. The antitumor effect of (-)-GA is more potent than that of its racemate (+)-GA [18-21]. However, (+)-gossypol has a stronger destructive effect on DNA bonds in normal human leukocytes than (-)-gossypol [17].

The differences in efficacy and adverse effects of gossypol optical isomers have been elucidated in previous studies[22]. To further clarify the mechanism of the effect of acetate gossypol optical isomers on uterine fibroids and liver injury, 1H-NMR technology was used to perform metabolomics analysis of rat serum to investigate the mechanism of gossypol optical isomers on uterine fibroids. It was found that gossypol could significantly improve the abnormality of tricarboxylic acid cycle, immune function, glycolysis and gluconeogenesis metabolism caused by uterine fibroids. Network toxicology was used to explore the mechanism of liver injury caused by acetate gossypol optical isomers. The potential targets of liver injury (HSP90AA1, HSP90AB1, SRC, MAPK1, AKT1, EGFR, BCL2, CASP3), and the molecular mechanisms of liver injury in rats (cancer pathway, PPAR signaling pathway, gluconeogenesis/glycolysis and Th17 cell differentiation) were elucidated. It is provided a research basis and reference for the further exploration of the drug use of optical isomers.

  1. The figure Quality must be improved, the font size in the figures is small and hardly visible.

Reply:

The quality of the images in this article has been improved and the size of the images has been modified.

Reviewer 3 Report

Comments and Suggestions for Authors

The authors have made the required revisions and the manuscript has been improved. For this reason, I now recommend it for publication.

Comments on the Quality of English Language

Minor editing of English language required.

Author Response

Response letter

Dear editors and reviewers,

We are grateful for your constructive comments and suggestions for our manuscript entitled “Using metabolomics to investigate the mechanism of action of optical isomers of gossypol acetate on uterine fibroids and predict the potential mechanism of the liver injury they cause using network toxicology” (ID: pharmaceuticals-3144089). Your comments are very valuable and helpful for improving our manuscript. In the following, the responses to all the comments are provided one by one.

We have tried our best to make all the revisions clear, and we hope that the revised manuscript can satisfy the requirements for publication.

The main revisions in the new manuscript are:

Reviewer 3

  1. The authors have made the required revisions and the manuscript has been improved. For this reason, I now recommend it for publication.

Reply:

We used MDPI's official website for English editing services.
